# Single Nucleotide Polymorphism and mRNA Expression of *LTF* in Oral Squamous Cell Carcinoma

**DOI:** 10.3390/genes13112085

**Published:** 2022-11-10

**Authors:** Karolina Gołąbek, Grzegorz Rączka, Jadwiga Gaździcka, Katarzyna Miśkiewicz-Orczyk, Natalia Zięba, Łukasz Krakowczyk, Dorota Hudy, Marek Asman, Maciej Misiołek, Joanna Katarzyna Strzelczyk

**Affiliations:** 1Department of Medical and Molecular Biology, Faculty of Medical Sciences in Zabrze, Medical University of Silesia, 19 Jordana Str., 41-808 Zabrze, Poland; 2Department of Forest Management Planning, Poznań University of Life Sciences, 71C Wojska Polskiego Str., 60-625 Poznan, Poland; 3Department of Otorhinolaryngology and Oncological Laryngology, Faculty of Medical Sciences in Zabrze, Medical University of Silesia in Katowice, 10 C Skłodowskiej Str., 41-800 Zabrze, Poland; 4Clinic of Oncological and Reconstructive Surgery, Maria Sklodowska-Curie National Research Institute of Oncology, 15 Wybrzeże Armii Krajowej Str., 44-102 Gliwice, Poland

**Keywords:** LTF, oral squamous cell carcinoma, single nucleotide polymorphism

## Abstract

Oral squamous cell carcinoma (OSCC) is one of the most prevalent types of cancers worldwide. LTF arrests the G1 to S phase transition of the cell cycle. This study is the first that has aimed to determine the possible association between the *LTF* polymorphisms (rs2073495, rs1126478, rs34827868, rs1042073, rs4637321, rs2239692 and rs10865941), the mRNA *LTF* expression, the risk of OSCC and the influence on the TNM staging and histological grading. This study was composed of 176 Polish patients, including 88 subjects diagnosed with OSCC and 88 healthy individuals. QuantStudio Design and Analysis Software v1.5.1 was used for the single nucleotide polymorphism (SNP) analysis and mRNA *LTF* expression. The G/G genotype of rs2073495 and the G/G genotype of rs4637321 were linked, with an increased risk of OSCC. There were no significant influences between the TNM staging and the histological grading and the *LTF* genotype. We found no statistically significant dissimilarities in the expression level of *LTF* genes in the tumour and margin specimens. No association was found between the gene expression levels, the other parameters or *LTF* polymorphisms in the tumour and margin samples. In conclusion, rs2073495 and rs4637321 polymorphisms may affect the risk of OSCC. These results should be validated on larger and different cohorts to better comprehend the role of the *LTF* gene in OSCC.

## 1. Introduction

Oral squamous cell carcinoma (OSCC) is one of the most prevalent types of cancers worldwide. In 2020, 377,713 new cases and 177,757 deaths were reported, with high incidence rates in Eastern and Western Europe [1]. The list of risk factors associated with OSCC includes the use of tobacco, alcohol consumption or human papillomavirus (HPV) infections [2]. Unfortunately, all causes of OSCC are not well-understood [3]. Several researchers have suggested that tobacco users have a better prognosis than non-smokers [4,5]. Unfortunately, there is no generally accepted method for screening for OSCC [6]. It seems, therefore, that another potential group of risk factors is related to the endogenous factors, such as a genetic predisposition. Single nucleotide polymorphisms (SNPs) are typical examples of this group [7].

Lactotransferrin, also known as lactoferrin (LTF, LF), is an iron-binding glycoprotein. Studies have shown that LTF has anticancer properties and suppresses the metastatic potential of cancer. LTF arrests the G1 to S phase transition of the cell cycle due to an induced expression and/or activity of critical cell cycle regulatory proteins, including Akt, p21, p19, p27, Cdk2, cyclin E, Cdk4 and cyclin D1 [8,9]. In addition, it has been shown that bovine LTF exerts a cytotoxic effect on fibrosarcoma and melanoma as well as head and neck and colon cancer cells and inhibits the proliferation of lung cancer cells [9]. It is also known for its antiparasitic, antibacterial, antifungal, antiviral and anti-oxidant properties as well as its immune regulatory activities [10].

This study is the first that has aimed to determine the possible association between the *LTF* polymorphisms (rs2073495, rs1126478, rs34827868, rs1042073, rs4637321, rs2239692 and rs10865941), the mRNA *LTF* expression, the risk of OSCC and the influence on the TNM staging and histological grading.

## 2. Materials and Methods

### 2.1. Patients and Samples

This study was composed of 176 Polish patients, including 88 subjects with OSCC and 88 healthy individuals (the control group). The tumour and control samples were described for the first time in a previous study [11]. We attempted to match the cases and controls for age, sex and sample size. All patients and controls were Caucasians who lived in Poland. The cancer samples were taken from Polish patients after surgical resections at the Department of Otorhinolaryngology and Oncological Laryngology of the Faculty of Medical Sciences in Zabrze at the Medical University of Silesia in Katowice and the Maria Sklodowska-Curie National Research Institute of Oncology (formerly known as the Maria Sklodowska-Curie Memorial Cancer Centre and Institute of Oncology), Gliwice, Poland. The tumour staging was based on the American Joint Committee on Cancer (AJCC, version 2007) [12,13] and the WHO Classification of Head and Neck Tumours [14]. Patients with an undiagnosed primary tumour and preoperative radiotherapy or chemotherapy were excluded from the study group. Buccal epithelial scrapings were collected from healthy individuals without a history of cancer at any site or potentially malignant disorders. The study was approved by the Bioethics Committee of the Medical University of Silesia (Katowice, Poland; approval no. KNW/022/KB1/49/16 and no. KNW/002/KB1/49/II/16/17) and the Institutional Review Board on Medical Ethics of the Maria Sklodowska-Curie Memorial Cancer Centre and Institute of Oncology, Gliwice, Poland (approval no. KB/493-15/08 and no. KB/430-47/13) [11].

### 2.2. LTF Polymorphism Collection

We collected seven SNPs based on a validated minimum 0.1 minor allele frequency (MAF) in the European population (National Center for Biotechnology Information, dbSNP) [15]; the potential relevance was confirmed by previous studies of other cancers [16,17]. These SNPs included rs2073495, rs1126478, rs34827868, rs1042073, rs4637321, rs2239692 and rs10865941. The characteristics of the polymorphisms used in the study are listed in Table 1.

### 2.3. DNA Isolation and SNP Genotyping

The methodology for the DNA isolation and the SNP genotyping was presented in a previous study [11]. Genomic DNA was extracted from each tumour sample (smaller than 20 mg) by DNeasy Blood & Tissue Kits (Qiagen, Hilden, Germany) according to the standard protocol. Before extraction, the samples were homogenised in a FastPrep^®^-24 instrument using Lysing Matrix A tubes (MP Biomedicals, Solon, CA, USA). In the healthy group, the DNA was extracted from the swabs taken from the buccal epithelial cells using a Swab-Extract DNA Purification Kit (EURx, Gdańsk, Poland) according to the manufacturer’s instructions. The qualitative and quantitative analyses of all isolated DNA were performed by spectrophotometry (NanoPhotometer Pearl, Implen, München, Germany).

The SNPs were genotyped with a QuantStudio 5 Real-Time PCR System and QuantStudio Design and Analysis Software v1.5.1 (Applied Biosystems, Foster City, CA, USA). The reaction solution contained 5 μg DNA (5.5 μL), 12.5 μL TaqMan Genotyping Master Mix (Applied Biosystems, Foster City, CA, USA) and 1.25 μL TaqMan Genotyping Assay (Applied Biosystems, Foster City, CA, USA). The assay IDs for the rs2073495, rs1126478, rs34827868, rs1042073, rs4637321, rs2239692 and rs10865941 polymorphisms were C__610621_1_, C__9698521_10, C__117160_10, C__2610629_1_, C__9106219_20, C__2610649_10 and C__357823_20, respectively. The cycle conditions were 95 °C for 10 min, 95 °C for 15 s and 60 °C for 1 min. The last two steps were repeated 40 times [11].

### 2.4. RNA Extraction and Gene Expression

The tumour and margin tissue samples were homogenised with a FastPrep^®^-24 homogenizer (MP Biomedicals, USA) using ceramic beads in Lysing Matrix D (MP Biomedicals, Solon, CA, USA). The RNA was isolated using an RNA isolation kit (BioVendor, Brno, Czech Republic) according to the standard instructions. The qualitative and quantitative analyses of all isolated RNA were performed by spectrophotometry in a Biochrom WPA Biowave DNA UV/Vis Spectrophotometer (Biochrom, Cambridge, UK). The total RNA (5 ng) was reverse-transcribed into cDNA using a High Capacity cDNA Reverse Transcription Kit with an RNase Inhibitor (Applied Biosystems, Foster City, CA, USA) according to the manufacturer’s protocol. The reaction was performed in a volume of 20 μL containing 2 μL of 10× Buffer RT, 0.8 μL of 25× dNTP mix (100 mM), 2 μL of 10× RT Random Primers, 1 μL of MultiScribe™ Reverse Transcriptase, 1 μL of RNase inhibitor, 3.2 μL of nuclease-free H_2_O and 10 μL of previously isolated RNA. The reaction was conducted in Mastercycler Personal Thermal Cyclers (Eppendorf, Hamburg, Germany) with the following thermal profiles: 25 °C for 10 min; 37 °C for 120 min; 85 °C for 5 min; and 4 °C –∞. The relative *LTF* gene expression analysis was performed by real-time PCR (qPCR) using TaqMan^TM^ Gene Expression Assays and QuantStudio 5 RealTime PCR System and Analysis Software v1.5.1 (Applied Biosystems, Foster City, CA, USA). The glyceraldehyde-3-phosphate dehydrogenase gene (*GAPDH*) was used as an endogenous control. The comparative threshold cycle (Ct) method 2^−∆∆Ct^ was used to determine the relative gene expression levels (relative quantification (RQ)). Seven surgical margin samples were used as a calibrator. The qPCR was performed in a volume of 20 µL using 1 µL of cDNA, 10 µL of TaqMan^TM^ Fast Advanced Master Mix (Applied Biosystems, Foster City, CA, USA), 1 µL of TaqMan^TM^ Gene Expression Assays (assay ID: Hs00914334_m1 for *LTF* and assay ID: Hs03929097_g1 for *GAPH*) and 8 µL of nuclease-free H_2_O (EURx, Gdańsk, Poland). The thermal cycles for all analysed genes were 95 °C for 20 s, followed by 40 cycles of 95 °C for 1 s and 60 °C for 20 s.

### 2.5. Statistical Analysis

The Hardy–Weinberg equilibrium was used separately for the control and cancer groups. Pearson’s χ^2^ test was used for the demographics and risk factor comparisons between the control and cancer groups. The significance between the distributions of genotypes and the clinical parameters was also tested using Pearson’s χ^2^ test. Odds ratios (ORs) and 95% confidence intervals (CIs) were obtained by Fisher’s exact test. The Kruskal–Wallis test was used to determine the expression of the *LTF* gene of the examined genotypes in the tumour and margin sections. Results with a *p*-value less than 0.05 were considered to be statistically significant. The statistical software STATISTICA version 13 (TIBCO Software Inc., Palo Alto, CA, USA) was used to perform all the analyses.

## 3. Results

### 3.1. Patient Characteristics

The clinical parameters of the OSCC group are shown in Table 2. In one patient, the histological grading (G) was not determined. The mean age was 56.4 years (range: 18–75 years). There were 60 (68.2%) men and 28 (31.8%) women; 69 (78.4%) patients were smokers, 65 (73.9%) reported alcohol consumption and 51 (57.9%) were both smokers and alcohol users. The mean age of the controls was 55 years (range: 40–87 years). This group comprised 44 (50%) men and 44 (50%) women, including 36 smokers (40.9%), 60 drinkers (68.2%); 34 individuals were both tobacco and alcohol users (38.6%).

### 3.2. Demographics and Risk Factors

In our study, the male gender (*p*-value < 0.001) and smoking (*p*-value < 0.001) were identified as the most important risk factors. There was no significant association between the OSCC risk and alcohol consumption or between concomitant alcohol consumption and cigarette smoking.

### 3.3. Distribution of LTF Genotypes

The *LTF* genotype frequencies are given in Table 3. All *LTF* genotypes followed the Hardy–Weinberg equilibrium, with the exception of rs2073495 and rs10865941. There was a significant difference in the distribution of the rs2073495, rs4637321 and rs10865941 genotypes between the OSCC and healthy individuals (*p*-value = 0.027, 0.033 and 0.002, respectively). The frequencies of the G/G variant of rs2073495 were higher in the cancer group than those in the healthy individuals. The homozygous variant A/A of rs4637321 and the homozygous variant T/T of rs10865941 were lower in the OSCC group than those in the healthy individuals.

### 3.4. LTF Genotypes and OSCC Risk

Table 4 shows the relationship between the *LTF* genotypes and OSCC risk. The G/G genotype of rs2073495 was associated with an increased OSCC risk (OR = 2.67; 95% CI = 1.20–5.91). A similar association was shown in one case of the G/G genotype of rs4637321 (OR = 6.25; 95% CI = 1.30–30.01).

### 3.5. LTF Genotypes and Clinicopathological Parameters

There were no significant associations between the clinicopathological parameters (such as the T, N and histological grading) and the *LTF* genotypes in the cancer group. The results are given in Table 5.

### 3.6. LTF mRNA Expression

We found no statistically significant differences in the relative expression level (RQ) of the *LTF* genes in the tumour samples compared with the margin samples (*p* = 0.378). The mean *LTF* relative gene expression in the tumour samples was 0.04 (SD = ±0.13); it was 0.13 (SD = ±0.29) in the margin samples. No association was found between the gene expression levels, age, gender, smoking, alcohol consumption, clinical parameters, TNM staging, histological grading and *LTF* polymorphisms in the tumour and margin samples.

## 4. Discussion

LTF protein plays an important role as a transcription factor and triggers the expression of a variety of genes, including genes related to the innate immune response, lipid metabolism, the inhibition of angiogenesis, apoptosis induction, DNA repair and cell cycle regulation [9]. There are only a few studies investigating the link between *LTF* gene polymorphisms and cancer risk. Zhou et al. [17] assessed the association of *LTF* gene polymorphisms (rs1126477, rs1126478, rs2073495 and rs9110) with nasopharyngeal carcinoma (NPC) in a Chinese population. According to Zhou et al., the NPC patients had a significantly higher frequency of the C allele of rs2073495 compared with the control group. This study showed that the CC genotype of rs2073495 was the risk factor for NPC. Moreover, the expression of the *LTF* gene was higher in the NPC tissues and control tissues with the A-G-G-T haplotype (constructed with rs1126477, rs1126478, rs2073495 and rs9110) compared with the samples without it [17]. *LTF* was downregulated in the NPC tissues, which was also observed in other studies [18,19]. The same polymorphisms were also analysed by Coa et al. [16]. It was found that the A-G-C-C haplotype (constructed with rs1126477, rs1126478, rs2073495 and rs9110) was a risk factor for ovarian cancer in the Chinese Han population. Moreover, the expression of the *LTF* gene was lower in patients with this haplotype [16]. Another study discovered that the G/G genotype of rs1126478 was correlated with an increased risk of chronic periodontitis [20]. In our study, the frequencies of the G/G variant of rs2073495 were higher in the OSCC group than those in the control group. In addition, the homozygous variants of A/A rs4637321 and of T/T of rs10865941 were lower in the OSCC group than those in the control group. The G/G genotype of rs2073495 was associated with an increased OSCC risk. A similar association was shown in one case of the G/G genotype of rs4637321. It can, therefore, be assumed that the rs2073495 and rs4637321 polymorphisms may affect the course of OSCC. However, further studies are warranted to confirm our findings because the small sample size decreased the statistical power.

A lower expression of *LTF* has been shown in tumours such as prostate cancer (CaP), NPC, osteosarcoma, clear cell renal cell carcinoma (ccRCC) and OSCC [21,22,23,24,25,26]. Shaheduzzaman et al. [21] demonstrated that *LTF* mRNA expression in tumour cells revealed a marked downregulation of *LTF*, with significant associations with a decreased prostate-specific antigen (PSA) recurrence-free survival of patients with CaP. LTF protein downregulation was observed both in tumour tissues and in serum. This suggests that LTF could be used for a cancer prognosis [21]. It is also known that *LTF* expression might be silenced by promoter hypermethylation in CaP [23]. Porter et al. [23] reported that *LTF* mRNA expression was silenced in prostate tumorigenesis via hypermethylation. The *LTF* CpG island is frequently and densely methylated in high-grade prostatic intraepithelial neoplasia, primary prostate carcinoma and metastases. On this basis, it can be assumed that changes in the methylation formula may be involved in the development of prostate cancer [23]. Chiu et al. [24] showed that *LTF* mRNA expression in ccRCC was significantly lower than in normal tissues. Changes in the expression of this gene indicated a poorer prognosis for the patients. A lower expression was associated with a poorer 5-year survival rate [24].

Zhang et al. [27] found that *LTF* mRNA expression was nearly undetectable in seven NPC cell lines [27]. Another study showed that *LTF* mRNA expression was significantly lower in NPC tissues than in normal tissues. Furthermore, *LTF* expression was negatively correlated with metastasis and the T stage. Of note, EBER-1 (EBV-encoded RNA 1) hybridisation signals were negatively correlated with *LTF* mRNA expression. In addition, changes in EBER-1 and *LTF* expression were significant risk factors for the development of NPC [28]. In our study, we found no statistically significant differences in the expression level of *LTF* genes in the tumour samples compared with the margin samples (*p*-value = 0.378). No association was found between the gene expression levels, age, gender, smoking status, alcohol consumption status, TNM staging, histological grading and *LTF* polymorphisms in the tumour and margin samples, which could mean that the rs2073495 and rs4637321 polymorphisms caused a difference in the stability and activity of the LTF protein in the absence of changes in the expression of the gene.

## 5. Conclusions

In summary, rs2073495 and rs4637321 polymorphisms may affect the risk of OSCC whereas rs1126478, rs34827868, rs1042073, rs2239692 and rs10865941 were not associated with OSCC. These results should be validated on larger and different cohorts to better comprehend role of the *LTF* gene in OSCC.

## Figures and Tables

**Table 1 genes-13-02085-t001:** Characteristics of *LTF* polymorphisms used in the study.

SNP ID	Location	SNP Type	Observed Codons	Observed Amino Acids
rs2073495	Chr.3: 46,439,467 (GRCh38)	Missense Mutation	GAC, GAG	D, E
rs1126478	Chr.3: 46,459,723 (GRCh38)	Missense Mutation	AAA, AGA	K, R
rs34827868	Chr.3: 46,452,365 (GRCh38)	Intron	–	–
rs1042073	Chr.3: 46,443,473 (GRCh38)	Silent Mutation	AAC, AAT	N, N
rs4637321	Chr.3: 46,465,324 (GRCh38)	Intron	–	–
rs2239692	Chr.3: 46,447,363 (GRCh38)	Silent Mutation	GGA, GGG	G, G
rs10865941	Chr.3: 46,467,057 (GRCh38)	Intron	–	–

**Table 2 genes-13-02085-t002:** Clinical parameters of patients with OSCC.

Clinical Parameters	Patients, n (%)
Histological Grading	
G1 (Well-differentiated)	14 (16)
G2 (Moderately differentiated)	60 (69)
G3 (Poorly differentiated)	13 (15)
T Classification	
T1	8 (9)
T2	24 (27.3)
T3	23 (26.2)
T4	33 (37.5)
Nodal Status	
N0	40 (45.5)
N1	25 (28.4)
N2	20 (22.7)
N3	3 (3.4)

**Table 3 genes-13-02085-t003:** Distribution of *LTF* genotypes in OSCC patients and healthy subjects.

SNP ID	Genotypes	OSCC n (%)	Control n (%)	*p*-Value
rs1126478	C/C	8 (9)	7 (8)	0.941
	C/T	32 (36)	34 (38)	
	T/T	48 (55)	48 (54)	
rs2073495	C/C	27 (31)	36 (41)	0.027
	C/G	29 (34)	37 (42)	
	G/G	30 (35)	15 (17)	
rs34827868	A/A	0 (0)	0 (0)	n.c.
	G/A	10 (11)	4 (5)	
	G/G	77 (89)	74 (95)	
rs1042073	A/A	6 (7)	8 (9)	0.728
	A/G	25 (30)	30 (34)	
	G/G	53 (63)	51 (57)	
rs4637321	A/A	2 (2)	10 (11)	0.033
	A/G	30 (35)	34 (39)	
	G/G	55 (63)	44 (50)	
rs2239692	C/C	0 (0)	5 (6)	n.c.
	C/T	12 (14)	3 (3)	
	T/T	76 (86)	81 (91)	
rs10865941	C/C	12 (14)	17 (19)	0.002
	C/T	54 (63)	33 (37)	
	T/T	20 (23)	39 (44)	

n.c.—not calculated; a *p*-value less than 0.05 was considered to be statistically significant.

**Table 4 genes-13-02085-t004:** Associations of *LTF* genotypes with odds ratios of OSCC patients and controls.

SNP ID	Genotypes	OSCC n (%)	Control n (%)	OR (95% CI)	*p*-Value
rs1126478	C/C	8 (9)	7 (8)	1	-
	C/T	32 (36)	34 (38)	0.82 (0.27–2.53)	>0.05
	T/T	48 (55)	48 (54)	0.87 (0.29–2.60)	>0.05
	C/T + T/T	80 (91)	82 (92)	0.85 (0.30–2.46)	>0.05
rs2073495	C/C	27 (31)	36 (41)	1	
	C/G	29 (34)	37 (42)	1.04 (0.52–2.10)	>0.05
	G/G	30 (35)	15 (17)	2.67 (1.20–5.91)	0.02
	C/G + G/G	59 (69)	52 (59)	1.51 (0.81–2.82)	>0.05
rs34827868	A/A	0 (0)	0 (0)	n.c.	n.c.
	G/A	10 (11)	4 (5)	n.c.	n.c.
	G/G	77 (89)	74 (95)	n.c.	n.c.
	G/A + G/G	87 (100)	78 (100)	n.c.	n.c.
rs1042073	A/A	6 (7)	8 (9)	1	
	A/G	25 (30)	30 (34)	1.11 (0.34–3.63)	>0.05
	G/G	53 (63)	51 (57)	1.39 (0.45–4.27)	>0.05
	A/G + G/G	78 (93)	81 (91)	1.28 (0.43–3.87)	>0.05
rs4637321	A/A	2 (2)	10 (11)	1	
	A/G	30 (35)	34 (39)	4.41 (0.89–21.76)	>0.05
	G/G	55 (63)	44 (50)	6.25 (1.30–30.01)	0.02
	A/G + G/G	85 (98)	78 (89)	5.45 (1.16–25.65)	0.03
rs2239692	C/C	0 (0)	5 (6)	n.c.	n.c.
	C/T	12 (14)	3 (3)	n.c.	n.c.
	T/T	76 (86)	81 (91)	n.c.	n.c.
	C/T + T/T	88 (100)	84 (94)	n.c.	n.c.
rs10865941	C/C	12 (14)	17 (19)	1	
	C/T	54 (63)	33 (37)	2.32 (0.98–5.46)	>0.05
	T/T	20 (23)	39 (44)	0.73 (0.29–1.81)	>0.05
	C/T + T/T	74 (86)	72 (81)	1.46 (0.65–3.26)	>0.05

n.c.—not calculated; a *p*-value less than 0.05 was considered to be statistically significant.

**Table 5 genes-13-02085-t005:** Association between *LTF* genotypes and T, N and histological grading in patients with OSCC.

			**T1**	**T2**	**T3**	**T4**	**N0**	**N1**	**N2**	**N3**	**G1**	**G2**	**G3**
rs1126478	C/C	n	1	1	4	2	3	3	2	0	1	5	2
		%	12.5	12.5	50.0	25.0	37.5	37.5	25.0	0.0	12.5	62.5	25.0
	C/T	n	4	11	7	10	15	10	5	2	6	19	6
		%	12.5	34.4	21.9	31.2	46.9	31.2	15.6	6.3	18.7	59.4	18.8
	T/T	n	3	12	12	21	22	12	13	1	7	36	5
		%	6.3	25.0	25.0	43.7	45.8	25.0	27.1	2.1	14.6	75.0	10.4
*p*-Value			0.492				0.903				0.565		
			**T1**	**T2**	**T3**	**T4**	**N0**	**N1**	**N2**	**N3**	**G1**	**G2**	**G3**
rs2073495	C/C	n	3	10	9	5	11	10	6	0	4	20	3
		%	11.1	37.1	33.3	18.5	40.7	37.1	22.2	0.0	14.8	74.1	11.1
	C/G	n	3	8	8	10	12	7	8	2	5	19	4
		%	10.3	27.6	27.6	34.5	41.4	24.1	27.6	6.9	17.2	65.5	13.8
	G/G	n	2	4	6	18	15	8	6	1	5	20	5
		%	6.7	13.3	20.0	60.0	50.0	26.7	20.0	3.3	16.6	66.7	16.7
*p*-Value			0.087				0.693				0.941		
			**T1**	**T2**	**T3**	**T4**	**N0**	**N1**	**N2**	**N3**	**G1**	**G2**	**G3**
rs34827868	A/A	n	0	0	0	0	0	0	0	0	0	0	0
		%	0.0	0.0	0.0	0.0	0.0	0.0	0.0	0.0	0.0	0.0	0.0
	G/A	n	2	1	3	4	4	4	1	1	1	6	3
		%	20.0	10.0	30.0	40.0	40.0	40.0	10.0	10.0	10.0	60.0	30.0
	G/G	n	6	23	19	29	36	21	18	2	13	54	9
		%	7.8	29.9	24.7	37.6	46.7	27.3	23.4	2.6	16.9	70.1	11.7
*p*-Value			0.421				0.438				0.269		
			**T1**	**T2**	**T3**	**T4**	**N0**	**N1**	**N2**	**N3**	**G1**	**G2**	**G3**
rs1042073	A/A	n	1	0	4	1	2	3	1	0	1	4	1
		%	16.7	0.0	66.7	16.6	33.3	50.0	16.7	0.0	16.7	66.7	16.6
	A/G	n	4	5	6	10	12	8	3	2	4	17	4
		%	16.0	20.0	24.0	40.0	48.0	32.0	12.0	8.0	16.0	68.0	16.0
	G/G	n	3	16	13	21	25	14	14	0	7	39	7
		%	5.7	30.2	24.5	39.6	47.2	26.4	26.4	0.0	13.2	73.6	13.2
*p*-Value			0.285				0.783				0.988		
			**T1**	**T2**	**T3**	**T4**	**N0**	**N1**	**N2**	**N3**	**G1**	**G2**	**G3**
rs4637321	A/A	n	0	2	0	0	1	1	0	0	0	2	0
		%	0.0	100.0	0.0	0.0	50.0	50.0	0.0	0.0	0.0	100.0	0.0
	A/G	n	4	6	9	11	14	10	6	0	4	21	5
		%	13.3	20.0	30.0	36.7	46.7	33.3	20.0	0.0	13.3	70.0	16.7
	G/G	n	4	16	14	21	25	14	13	3	10	36	8
		%	7.3	29.1	25.4	38.2	45.5	25.4	23.6	5.5	18.2	65.5	14.5
*p*-Value			0.674				0.591				0.739		
			**T1**	**T2**	**T3**	**T4**	**N0**	**N1**	**N2**	**N3**	**G1**	**G2**	**G3**
rs2239692	C/C	n	0	0	0	0	0	0	0	0	0	0	0
		%	0.0	0.0	0.0	0.0	0.0	0.0	0.0	0.0	0.0	0.0	0.0
	C/T	n	3	0	5	4	6	4	1	1	2	8	2
		%	25.0	0.0	41.7	33.3	50.0	33.3	8.3	8.4	16.6	66.7	16.7
	T/T	n	5	24	18	29	34	21	19	2	12	52	11
		%	6.6	31.6	23.7	38.1	44.8	27.6	25.0	2.6	15.8	68.4	14.5
*p*-Value			0.398				0.487				0.98		
			**T1**	**T2**	**T3**	**T4**	**N0**	**N1**	**N2**	**N3**	**G1**	**G2**	**G3**
rs10865941	C/C	n	1	4	4	3	7	3	2	0	3	8	1
		%	8.3	33.4	33.3	25.0	58.3	25.0	16.7	0.0	25.0	66.7	8.3
	C/T	n	5	15	12	22	24	15	14	1	9	37	7
		%	9.3	27.8	22.2	40.7	44.4	27.8	25.9	1.9	16.7	68.5	13.0
	T/T	n	2	4	7	7	8	7	3	2	2	14	4
		%	10.0	20.0	35.0	35.0	40.0	35.0	15.0	10.0	10.0	70.0	20.0
*p*-Value			0.881				0.844				0.752		

A *p*-value less than 0.05 was considered to be statistically significant.

## Data Availability

The data used to support the findings of this study are available from the corresponding author upon request.

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
