# Peer review of "Single Nucleotide Polymorphism and mRNA Expression of LTF in Oral Squamous Cell Carcinoma"

_genes, 2022, doi:10.3390/genes13112085_

Round 1

Reviewer 1 Report

(Manuscript ID: genes-2002929) concerns studies of possible effect LTF gene polymorphism has on risk of development of oral squamous cell carcinoma (OSCC). The authors investigated the frequency of seven polymorphic variants of this gene in the groups of patients and healthy donors. The groups were not very numerous, but well selected and characterized in terms of gender and age. The methods are chosen properly and described clearly except of genotyping SNPs where the catalogue numbers (such as C___2610621_1_) seem to be not proper (p.3 line 101-102). In Materials and Methods it is not mentioned whether polymorphic variants were tested in DNA of healthy cells or exclusively isolated from neoplastic cells (theoretically, it is possible that the carcinogenesis accelerates the emergence of certain polymorphic variants). Results are presented mainly in tables however some tables would need additional information i.e. in tables 3 and 4 the last column described as p should be explained. Also Table 4 does not show odds but odds ratio. The results show that distribution of rs2073495, rs4637321 and rs10865941 variant genotypes differs between healthy and OSCC groups and the homozygous genotypes of first and second form increase the chance of OSCC whereas the none of genotypes of third polymorphism significantly influence OSCC risk. Table 5 is to show whether some polymorphic variants are not conducive to OSCC growth, metastasis or differentiation but again reader does not know what is p and how it was calculated. The last paragraph of the results on LTF mRNA expression doesn't make much sense with regard to the polymorphic variants. The authors compare tumor RNA expression with expression within the theoretically healthy margin, but this is not very much related to the LTF polymorphism. Most likely, the LTF gene in the tumor will contain the same polymorphic variants as the margin. Discussion shows some literature data however not all is clear, i.e. what means ‘A-G-G-T’ haplotype, it should be explained which variant in used in manuscript annotation it represents. In respect to their own results the Authors write in Discussion that G/G genotype of rs2073495 was associated with decreased OSCC risk (page 8, line 200) but results presented in Table 4 show opposite. The authors devote a lot of space in the Discussion to the level of mRNA expression, citing different values of expression obtained by others for different types of cancer, but it seems that this part of the discussion is redundant because it is not related to polymorphism and does not bring valuable information to the manuscript. In the Discussion, the authors repeat the description of the results (not quite exactly), but not even any suggestions about the mechanism of LTF operation.

Author Response

We are grateful to the Reviewer for constructive and helpful comments. Please see the attachment

Reviewer 2 Report

The authors presented the "Single nucleotide polymorphism and mRNA expression of LTF in oral squamous cell carcinoma". I found the study interesting with novel findings. However, it could have been better to study couple more cohorts and design more graphically informative figures. 

Author Response

We are grateful to the Reviewer for constructive comments. Please see the attachment.

Reviewer 3 Report

Comment to Author

The author attempt to identify a prognostic marker for OSCC, though they have missed several key factors to consider in the present manuscript.

1. The aim of the study is introduced in a way that apart from smoking, HPV, or alcohol, other factors may contribute to the OSCC prognosis, which is important and reasonable (OSCC risk in non-smokers) but the authors did not provide any clear literature or data to support their study. Lines 150-153 stated "gender and smoking are significant risk factors" in the present study followed by "no significant association between OSCC risk and alcohol consumption, or between alcohol consumption and cigarette smoking".  This needs to be discussed clearly. 

2. The present study shows that G/G genotypes rs2073495 and rs4637321 are associated with decreased OSCC risk (line, 166-168) and there is no association among the progression of tumor (clinicopathological parameters). This needs to be addressed with literature support for better understanding. 

3. The buccal swabs were taken from healthy control to consider control subjects for the study. The majority of the tumor samples are from the mandible and tongue (from Ref. 11, authors' previous paper) which is hard to compare. Moreover, the authors did not mention whether all the control samples matched with age, sex, or ethnicity. which is the significant criterion followed generally. 

4. The rationale behind assessing LTF mRNA expression is not clear, the mean expression level is 0.04 (line 181) compared to what? which is not clear to understand (generally gene expression is presented as fold change using 2power delta delta ct or another clear format) and there is no histogram or other format data, primer sequence is provided with the manuscript. 

4. As a minor change, authors can bold significance (P value) in tables 3 & 4 for better visibility; line adjustment in table 4 for (n) and %; line 62 sentence did not complete, and space adjustments need to be done throughout the manuscript. 

Author Response

We are grateful to the Reviewer for constructive and helpful comments. Please see the attachment.

Round 2

Reviewer 3 Report

The authors did explain to my comments